# A novel ciprofloxacin-resistant subclade of H58 *Salmonella* Typhi is associated with fluoroquinolone treatment failure

Duy Pham Thanh[1], Abhilasha Karkey[2], Sabina Dongol[2], Nhan Ho Thi[1], Corinne N Thompson[1,3,4], Maia A Rabaa[1,3], Amit Arjyal[2], Kathryn E Holt[5], Vanessa Wong[6], Nga Tran Vu Thieu[1], Phat Voong Vinh[1], Tuyen Ha Thanh[1], Ashish Pradhan[7], Saroj Kumar Shrestha[7], Damoder Gajurel[7], Derek Pickard[6], Christopher M Parry[4,8], Gordon Dougan[6], Marcel Wolbers[1,3], Christiane Dolecek[1,3,9], Guy E Thwaites[1,3], Buddha Basnyat[2], Stephen Baker[1,3,4]*

[1]The Hospital for Tropical Diseases, Wellcome Trust Major Overseas Programme, Oxford University Clinical Research Unit, Ho Chi Minh City, Vietnam; [2]Oxford University Clinical Research Unit, Patan Academy of Health Sciences, Kathmandu, Nepal; [3]Centre for Tropical Medicine and Global Health, Oxford University, Oxford, United Kingdom; [4]The London School of Hygiene and Tropical Medicine, London, United Kingdom; [5]Department of Biochemistry and Molecular Biology, Bio21 Molecular Science and Biotechnology Institute, The University of Melbourne, Melbourne, Australia; [6]The Wellcome Trust Sanger Institute, Cambridge, United Kingdom; [7]Civil Services Hospital, Kathmandu, Nepal; [8]School of Tropical Medicine and Global Health, Department of Clinical Research, Nagasaki University, Nagasaki, Japan; [9]Department of Zoology, University of Oxford, Oxford, United Kingdom

*For correspondence: sbaker@oucru.org

**Competing interests:** The authors declare that no competing interests exist.

**Abstract** The interplay between bacterial antimicrobial susceptibility, phylogenetics and patient outcome is poorly understood. During a typhoid clinical treatment trial in Nepal, we observed several treatment failures and isolated highly fluoroquinolone-resistant *Salmonella* Typhi (*S.* Typhi). Seventy-eight *S.* Typhi isolates were genome sequenced and clinical observations, treatment failures and fever clearance times (FCTs) were stratified by lineage. Most fluoroquinolone-resistant *S.* Typhi belonged to a specific H58 subclade. Treatment failure with *S.* Typhi-H58 was significantly less frequent with ceftriaxone (3/31; 9.7%) than gatifloxacin (15/34; 44.1%)(Hazard Ratio 0.19, p=0.002). Further, for gatifloxacin-treated patients, those infected with fluoroquinolone-resistant organisms had significantly higher median FCTs (8.2 days) than those infected with susceptible (2.96) or intermediately resistant organisms (4.01)(p<0.001). H58 is the dominant *S.* Typhi clade internationally, but there are no data regarding disease outcome with this organism. We report an emergent new subclade of *S.* Typhi-H58 that is associated with fluoroquinolone treatment failure. Clinical trial registration: ISRCTN63006567.

## Introduction

Enteric (typhoid) fever, a systemic infection caused predominantly by the bacterium *Salmonella enterica* subspecies *enterica* serovar Typhi (*S.* Typhi), remains one of the principal bacterial causes of febrile disease in low-income countries (*Parry et al., 2002*). *S.* Typhi is a distinct, monophyletic lineage of *S. enterica* that is exquisitely adapted to cause disease only in humans (*Roumagnac et al., 2006*), characterised by a non-specific fever with malaise and asymptomatic convalescent carriage

**eLife digest** People who ingest a type of bacteria called *Salmonella* Typhi can develop the symptoms of typhoid fever. This disease is common in low-income settings in Asia and Africa, and causes a high rate of death in people who are not treated with antimicrobial drugs.

During a study in Nepal, Thanh et al. tried to evaluate which of two antimicrobials was better for treating typhoid fever. One of the drugs – called gatifloxacin – did not work in some of the patients. To understand why this treatment failed, Thanh et al. decoded the entire DNA sequences of all the *Salmonella* Typhi bacteria isolated during the study. Comparing this genetic data to the clinical data of the patients identified a new variant of *Salmonella* Typhi. These bacteria have a specific combination of genetic mutations that render them resistant to the family of drugs that gatifloxacin belongs to – the fluoroquinolones.

Patients infected with the variant bacteria and treated with gatifloxacin were highly likely to completely fail treatment and have longer-lasting fevers. On further investigation Thanh et al. found these organisms were likely recently introduced into Nepal from India.

Fluoroquinolones are amongst the most effective and common antimicrobials used to treat typhoid fever and other bacterial infections. However, the presence of bacteria that are resistant to these compounds in South Asia means that they should no longer be the first choice of drug to treat typhoid fever in this location.

(*Parry et al., 2002*). There are an estimated 20–30 million new cases of enteric fever per year globally (*Crump and Mintz, 2010*), with the majority occurring in Asia, but there is an increasingly recognised burden of disease across sub-Saharan Africa.

Antimicrobial resistance is a major global health challenge, and resistance against the most commonly used antimicrobials for treating enteric fever has evolved successively over the last 30 years. Ampicillin, chloramphenicol and trimethoprim-sulfamethoxazole were originally standard-of-care for enteric fever. However, multidrug resistance (MDR) against these agents began to emerge in the 1970s and 1980s (*Olarte and Galindo, 1973*; *Wain et al., 2003*). Consequently, third-generation cephalosporins and fluoroquinolones (FQs) became the most clinically reliable drugs for treating enteric fever (*Kariuki et al., 2015*), and were formally advocated by the World Health Organization (WHO) in 2003 (*World Health Organization, 2003*). *S.* Typhi isolates with acquired resistance against third-generation cephalosporins are rare (*Hendriksen et al., 2015*), but *S.* Typhi exhibiting reduced susceptibility to FQs, induced by sequential mutations in the gene encoding a target protein (*gyrA*), now dominate internationally (*Emary et al., 2012*; *Kariuki et al., 2010*). The global ascendency of *S.* Typhi strains with reduced susceptibility to FQs has been partly catalysed by the dissemination of a specific MDR lineage (H58) across Asia and Africa (*Wong et al., 2015*). These H58 strains are rapidly displacing other lineages, and strains with *gyrA* mutations may have a fitness advantage, even in the absence of antimicrobial exposure (*Baker et al., 2013*).

We have previously shown that protracted fever clearance times (FCTs) are associated with organisms with higher Minimum Inhibitory Concentrations (MIC) against FQs in enteric fever patients treated with ciprofloxacin and ofloxacin (*Parry et al., 2011*). However, whilst the clinical efficacy of the older FQs in enteric fever is contentious, we have shown that the fourth-generation FQ, gatifloxacin, has remained efficacious for uncomplicated disease, even in patients infected with *S.* Typhi strains with reduced ciprofloxacin susceptibility (MIC $\geq$0.125 µg/mL) (*Pandit et al., 2007*; *Koirala et al., 2013*; *Arjyal et al., 2011*).

During a recent randomised controlled trial (RCT) comparing ceftriaxone and gatifloxacin, conducted in Nepal, we observed an increased number of treatment failures associated with FQ-resistant (ciprofloxacin MIC>32 µg/ml) *S.* Typhi, prompting the data safety and monitoring board to stop the trial (*Arjyal et al., 2016*). Aiming to assess the molecular epidemiology of the infecting isolates and investigate how genotype may be related to treatment outcome, we performed whole genome sequencing (WGS) on the *S.* Typhi isolated during this trial, and after stratifying by genotype, we assessed clinical presentation and outcome.

## Results

### *Salmonella* Typhi whole genome sequencing

We performed WGS on the 78 available *S.* Typhi isolates from patients in both RCT treatment arms (gatifloxacin and ceftriaxone) (*Supplementary file 1*). The resulting phylogeny, which incorporated reference sequence CT18, indicated that the majority of isolates (65/78; 83.3%) fell within the H58 lineage, while the remaining 13 (16.7%) represented eight different lineages (*Figure 1*). All but four of the H58 strains contained the common DNA gyrase (*gyrA*) mutation in codon 83 (S83F), which confers reduced susceptibility to FQs (ciprofloxacin MIC; 0.125–0 5 µg/ml) (*Parry et al., 2010*). Nested within the S83F H58 group, but separated from the rest of the group by a branch defined by 30 SNPs, was an H58 subclade comprised of 12 isolates containing the S83F *gyrA* mutation, a mutation in *gyrA* at codon 87 (D87N), and an additional mutation in the topoisomerase gene, *parC* (S80I) (H58 triple mutant). Notably, these H58 triple mutants shared high MICs against ciprofloxacin (≥24 µg/ml). Further, an additional two non-H58 RCT isolates with ciprofloxacin MIC≥24 µg/ml had the S83F *gyrA* mutation, an alternative mutation at codon 87 (D87V), the S80I *parC* mutation, and an A364V mutation in *parE* (*Figure 1*, *Supplementary file 1*). Notably, none of the sequenced isolates harboured plasmid-mediated quinolone resistance genes (PMQR) or contained additional antimicrobial resistance genes within the well-described *S.* Typhi-associated IncH1 family of plasmids.

### Clinical presentation of *Salmonella* Typhi infections

We stratified clinical data from the RCT by H58 status of the corresponding *S.* Typhi isolates (H58; N=65, non-H58; N=13) and compared baseline characteristics between these groups. We found no significant differences in demographics and no association between disease severity at presentation

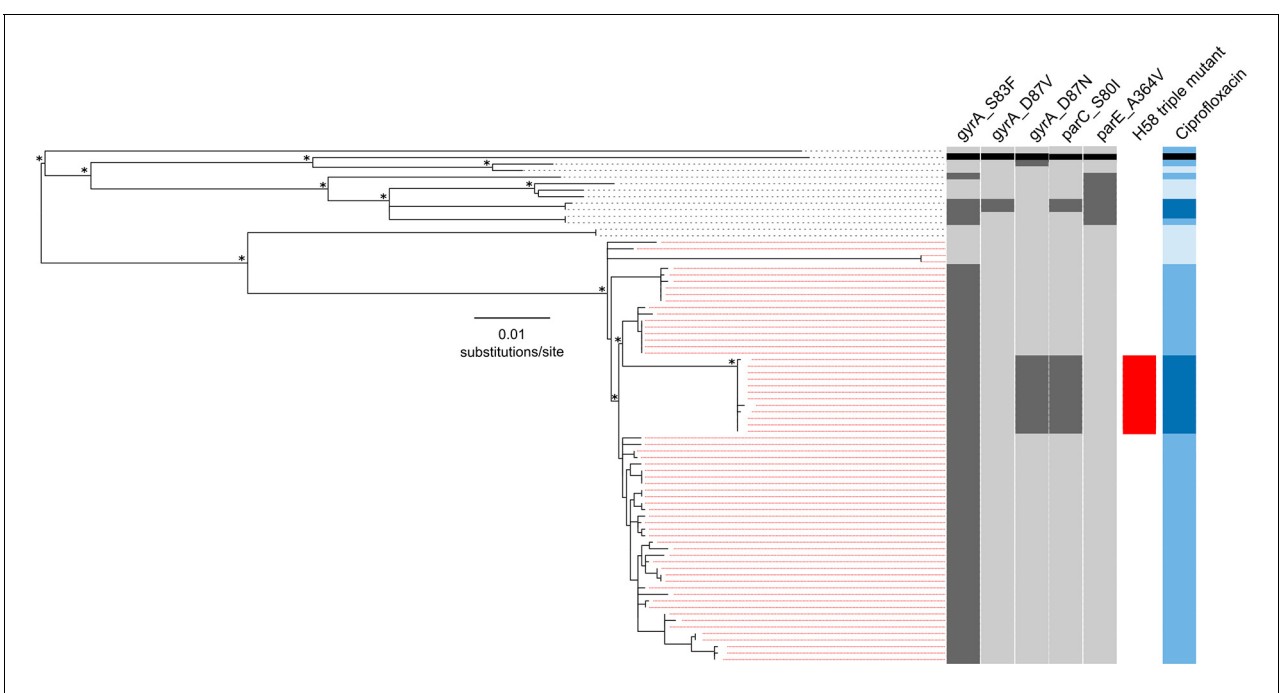

**Figure 1.** The phylogenetic structure of 78 Nepali *Salmonella* Typhi isolated during a gatifloxacin versus ceftriaxone randomised controlled trial. Maximum likelihood phylogeny based on core-genome SNPs of 78 *Salmonella* Typhi RCT isolates with the corresponding metadata, including the presence of mutations (dark grey) in *gyrA* (S83F, D87V and D87N), *parC* (S80I) and *parC* (A364V) and susceptibility to ciprofloxacin (susceptible, light blue; intermediate, mid-blue and non-susceptible, dark blue) by Minimum Inhibitory Concentration (MIC). The reference strain CT18 was used for context and highlighted by the black boxes. Red lines linking to metadata show isolates belonging to the *Salmonella* Typhi H58 lineage (with H58 triple mutants highlighted), other lineages (non-H58) are shown with black lines. The scale bar indicates the number of substitutions per variable site (see methods). Asterisks indicate ≥85% bootstrap support at nodes of interest.

between those infected with an H58 *S.* Typhi isolate or a non-H58 isolate (*Supplementary file 2A*). Next, we compared the baseline characteristics of patients stratified by ciprofloxacin susceptibility (susceptible, intermediate and resistant), and found no differences in disease severity or demographics on presentation; the only exception being that FQ-resistant *S.* Typhi were more frequently isolated from adults (*Supplementary file 2B*). A significantly lower proportion of H58 *S.* Typhi (4/65; 6.2%) were susceptible to FQs compared to non-H58 isolates (6/13; 46%) (p=0.001) (*Table 1*) and, overall, H58 isolates had significantly higher (but not resistant) MICs against the majority of tested antimicrobials than non-H58 isolates (*Table 1*).

## Treatment failure and fever clearance times

The primary endpoint of the RCT in which these data were generated was a composite for treatment failure (see method and previous publication) (*Arjyal et al., 2016*). Treatment failure with H58 *S.* Typhi was significantly less common in the ceftriaxone group (3/31; 9.7%) than the gatifloxacin group (15/34; 44.1%) (Hazard Ratio (HR) of time to failure 0.19, 95%CI 0.05–0.56, p=0.002) (*Table 2*). Conversely, there was no significant difference in treatment failure between those infected with non-H58 isolates treated with gatifloxacin (0/6; 0%) or ceftriaxone (2/7; 28.6%) (p=0.32). Similarly, time to fever clearance differed significantly between the two treatment groups in H58 infections, with median FCTs of 5.03 days (interquartile range (IQR): 3.18–7.21) in the gatifloxacin group and 3.07 days (IQR: 1.89–4.52) in the ceftriaxone group (p<0.0006). Again, this trend was not mirrored in the non-H58 *S.* Typhi infections, with FCTs of 2.87 (IQR: 2.08–3.7) and 3.12 (IQR: 2.2–4.12) days for gatifloxacin and ceftriaxone, respectively (p=0.61) (*Table 3*). Moreover, in the gatifloxacin arm, H58 *S.* Typhi tended to be associated with a higher risk of treatment failure (p=0.06) and a longer fever clearance time (p=0.013) (*Figure 2*, *Table 2* and *Supplementary file 2C*).

As we identified two non-H58 isolates that were also FQ-resistant (*Figure 1*), we additionally stratified outcome for the gatifloxacin arm (N=40 patients) by FQ susceptibility of the infecting organism. Those infected with FQ-resistant *S.* Typhi failed gatifloxacin treatment more frequently (8/10; 80%) than those infected with an intermediately resistant organism (7/25; 28%) or a susceptible organism (0/5; 0%) (p=0.007) (*Figure 2* and *Table 2*). Furthermore, in the gatifloxacin arm, those infected with FQ-resistant organisms had significantly higher median FCTs than those infected with

**Table 1.** Comparison of antimicrobial susceptibility by *Salmonella* Typhi lineage.

| E test | Non-H58 (N=13) | | | H58 (N=65) | | | p value* |
|---|---|---|---|---|---|---|---|
| | MIC50 | MIC90 | GM (range) | MIC50 | MIC90 | GM (range) | |
| Amoxicillin | 0.5 | 1 | 0.77 (0.38–38) | 0.75 | >256 | 1.43 (0.38–>256) | 0.0412 |
| Chloramphenicol | 3 | 4 | 2.7 (1.5–8) | 4 | 12 | 5.7 (2–>256) | 0.0147 |
| Ceftriaxone | 0.06 | 0.06 | 0.06 (0.05–0.13) | 0.09 | 0.19 | 0.11 (0.03–0.64) | 0.0004 |
| Gatifloxacin | 0.13 | 0.25 | 0.06 (0.01–2) | 0.13 | 2 | 0.21 (0.01–3) | 0.1197 |
| Nalidixic acid | >256 | >256 | 21.6 (1–>256) | >256 | >256 | 346.8 (1–>256) | 0.0004 |
| Ofloxacin | 0.25 | 0.75 | 0.24 (0.03–>32) | 0.5 | >3232 | 1.09 (0.03–>32) | 0.0240 |
| Trimethoprim sulphate | 0.02 | 0.05 | 0.03 (0.02–0.05) | 0.05 | 0.32 | 0.09 (0.01–>32) | 0.0016 |
| Ciprofloxacin | 0.13 | 0.75 | 0.11 (0.01–>32) | 0.38 | >32 | 0.80 (0.02–>32) | 0.0051 |
| Ciprofloxacin susceptibility group | | | | | | | 0.0008# |
| - Susceptible | 6 (46.2%) | | | 4 (6.2%) | | | |
| - Intermediate | 4 (30.8%) | | | 48 (73.8%) | | | |
| - Resistant | 3 (23.1%) | | | 13 (20.0%) | | | |

*Comparisons between *Salmonella* Typhi lineage for MICs and ciprofloxacin susceptibility groups were based on the Wilcoxon rank sum test and Fisher's exact test. respectively.

MIC: minimum inhibitory concentration, measured in μg/ml

#p value for comparison of susceptible vs. intermediate/resistant combined between groups by Fisher's exact test is 0.001.

GM: geometric mean, the upper range of the values was determined by multiplying the MIC by 2 if the result was >X (for example, >256 = 256*2 = 512).

**Table 2.** Summary of time to treatment failure by *Salmonella* Typhi lineage and ciprofloxacin susceptibility.

| Time to treatment failure | Gatifloxacin (events/N) | Ceftriaxone (events/N) | Hazard ratio of time to failure (95%CI); p value | Heterogeneity test (p value) |
|---|---|---|---|---|
| H58* | | | | 0.020 |
| - H58 | 15/34 | 3/31 | 0.19 (0.05, 0.56); p=0.002 | |
| - Non-H58 | 0/6 | 2/7 | 3.87 (0.31, 534.24); p=0.32 | |
| Ciprofloxacin susceptibility group† | | | | 0.08 |
| - Susceptible | 0/5 | 1/5 | 2.40 (0.13, 350.21); p=0.57 | |
| - Intermediate | 7/25 | 2/27 | 0.27 (0.05, 0.99); p=0.049 | |
| - Resistant | 8/10 | 2/6 | 0.27 (0.05, 1.01); p=0.052 | |

*Likelihood ratio test p=0.06 and 0.40 for comparison of time to treatment failure between H58 vs. non-H58 groups in gatifloxacin arm only and in all patients, respectively

†Likelihood ratio test p=0.007 for comparison of time to treatment failure between MIC groups in gatifloxacin arm only

*S.* Typhi with alternative FQ susceptibility profiles (median FCTs (days): susceptible, 2.96 (IQR: 2.13–3.85), intermediate, 4.01 (IQR: 2.76–5.37) and resistant 8.2 (IQR: 5.99–10.5), respectively [p<0.0001]) (*Table 3* and *Supplementary file 2D*). Comparatively, the median FCT for those infected with an FQ-resistant organism but randomised to ceftriaxone was 3.83 days (IQR: 2.96–4.7) (p<0.0001 for the between-treatment comparison).

## The emergence of fluoroquinolone-resistant *Salmonella* Typhi

To measure the pattern of emergence of FQ-resistant *S.* Typhi in Nepal, we compiled FQ susceptibility data from 837 organisms isolated during enteric fever RCTs conducted at Patan Hospital between 2005 and 2014 (*Figure 3*) (*Pandit et al., 2007*; *Koirala et al., 2013*; *Arjyal et al., 2011*). MICs against FQs were generally higher for *S.* Paratyphi A than for *S.* Typhi. There was a significant temporal increase in *S.* Typhi MICs against both ciprofloxacin (p<0.0001) and gatifloxacin (p<0.0001), with a sharp increase from 2009. MICs against gatifloxacin in *S.* Paratyphi A also significantly increased with time (p<0.0001); however, MICs against ciprofloxacin showed only weak evidence of an upward trend over time (p=0.06).

We hypothesised that the H58 triple mutants represented a contemporary importation into Nepal. To explore this, we compared the genomes of the 78 RCT *S.* Typhi isolates with those from 58 supplementary *S.* Typhi isolates from previous studies conducted between 2008 and 2013 in this setting (*Figure 4*, *Supplementary file 1*) (*Wong et al., 2015*). We found that the majority of the local H58 isolates (84/121; 69.4%) were closely related; these strains represented an 'endemic' Nepali H58 clade containing a single S83F *gyrA* mutation. Additionally, we identified a further five Nepali strains isolated in 2013 that belonged to the H58 triple mutant group, and had an MIC ≥24 μg/ml against ciprofloxacin. Incorporating additional genome sequences from a recent international study of the H58 lineage (*Wong et al., 2015*), we found that all the Nepali H58 triple mutants were very closely related (5 SNPs to nearest neighbour) to H58 triple mutants isolated previously in neighbouring India between 2008 and 2012 (*Figure 4*).

## Discussion

Our study shows that a new FQ-resistant subclade of H58 *S.* Typhi has been introduced into Nepal and is associated with a lack of FQ efficacy. This subclade was associated with longer FCTs and treatment failure in patients treated with the FQ, gatifloxacin. For the first time, we can conclusively show how enteric fever patients respond to FQ treatment when infected with a specific subclade of H58, thereby linking organism genotype with a treatment phenotype. Given the international significance of FQs for the treatment of enteric fever and other bacterial infections, our findings have major global health implications for the long-term use and efficacy of this group of antimicrobials.

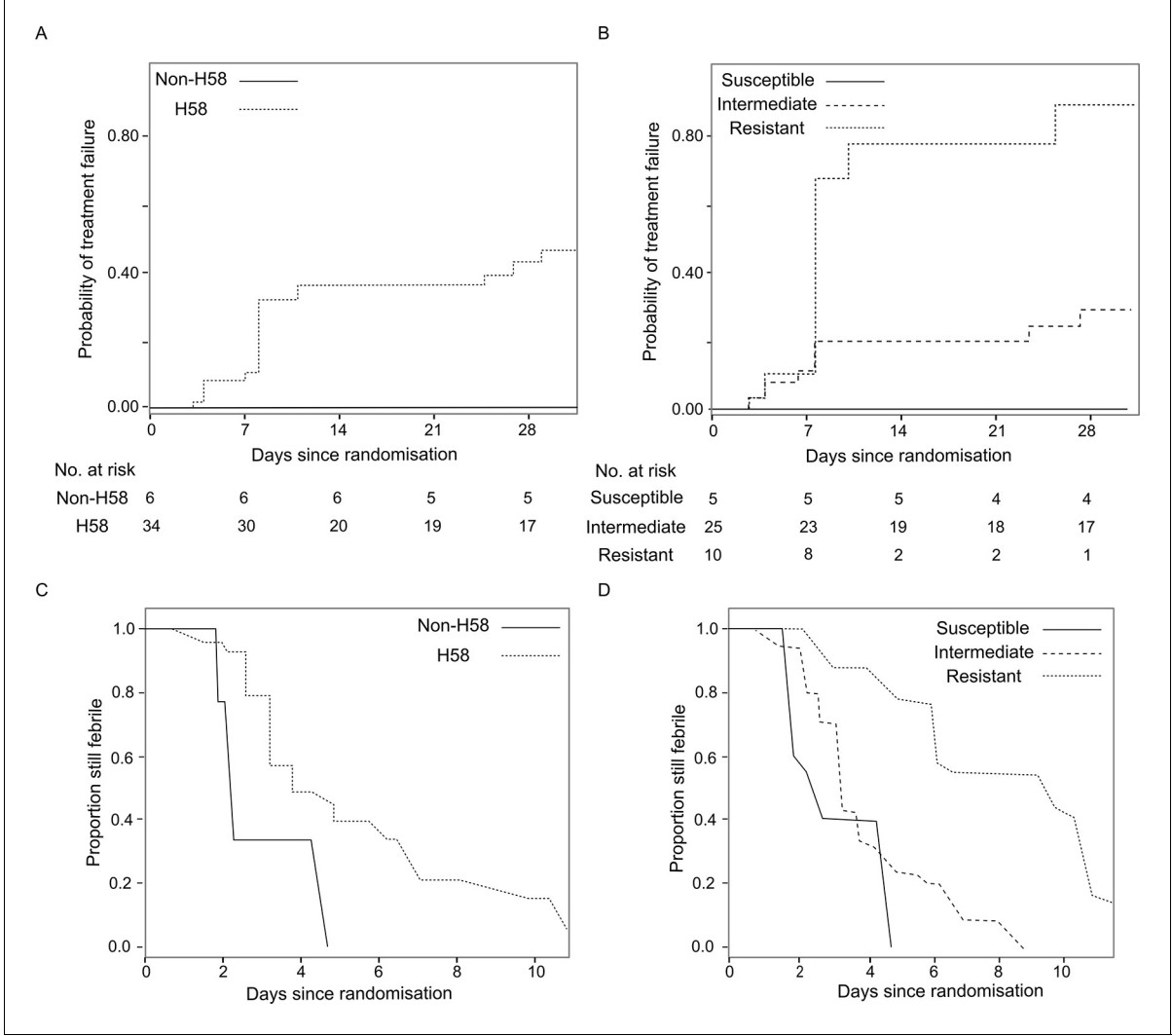

**Figure 2.** The association of *Salmonella* Typhi lineage and ciprofloxacin susceptibility with treatment failure and fever clearance time in patients randomised to gatifloxacin. (**A**) Kaplan-Meier curve for time to treatment failure by H58 and non-H58 *Salmonella* Typhi. (**B**) Kaplan-Meier curve for time to treatment failure by *Salmonella* Typhi susceptibility group (susceptible, intermediate, resistant to ciprofloxacin). (**C**) Non-parametric maximum likelihood estimators for interval-censored fever clearance time (see methods) by H58 and non-H58 *Salmonella* Typhi. (**D**) Non-parametric maximum likelihood estimators for interval-censored fever clearance time by *Salmonella* Typhi susceptibility group (susceptible, intermediate, resistant to ciprofloxacin).

Our data suggest these FQ-resistant *S.* Typhi strains circulating in Nepal most likely descended from a single ancestor carrying the triple *gyrA/parC* mutant, such as that isolated in Nepal in 2011 (***Koirala et al., 2012***). This isolate was also associated with treatment failure, although this organism was not genome sequenced and was assumed to be an isolated case. More significantly, several very closely related strains were genome sequenced during an international study of H58 *S.* Typhi (***Wong et al., 2015***). These organisms had the same combination of triple FQ resistance mutations as those described here; our analysis shows they belong to the same subclade of H58. These strains had equivalently high MICs against ciprofloxacin and were isolated in India between 2008 and 2012. However, there were no associated patient outcome data for these strains and other reports from India have been limited. Our data implies that this lineage was introduced into Nepal from India or elsewhere in South Asia within the last 4–5 years and has subsequently entered in an endemic transmission cycle in Kathmandu. Given the large extent of human movement between India and Nepal, we propose this is the most likely route of introduction. However, there is also a small possibility

**Table 3.** Summary of fever clearance time by Salmonella Typhi lineage and ciprofloxacin susceptibility.

| Fever clearance time | Gatifloxacin median (IQR) days | Ceftriaxone median (IQR) days | Acceleration factor (95%CI); p value | Heterogeneity test (p value) |
|---|---|---|---|---|
| H58[¥] | | | | 0.07 |
| - H58 | 5.03 (3.18, 7.21) | 3.07 (1.89 ,4.52) | 1.59 (1.22, 2.09); p=0.0006 | |
| - Non-H58 | 2.87 (2.08, 3.7) | 3.12 (2.2, 4.12) | 0.90 (0.59, 1.36); p=0.61 | |
| Ciprofloxacin susceptibility group[‡] | | | | 0.015 |
| - Susceptible | 2.96 (2.13, 3.85) | 4.78 (4.01, 5.5) | 0.71 (0.49, 1.02); p=0.07 | |
| - Intermediate | 4.01 (2.76, 5.37) | 2.63 (1.52, 4.05) | 1.31 (0.97, 1.76); p=0.07 | |
| - Resistant | 8.2 (5.99, 10.5) | 3.83 (2.96, 4.7) | 2.23 (1.57, 3.17); p<0.0001 | |

[¥]p=0.013 and p=0.029 for comparison of interval censored time to fever clearance between H58 vs. non-H58 groups in gatifloxacin arm only and in all patients, respectively

[‡]p<0.0001 for comparison of interval censored time to fever clearance between MIC groups in gatifloxacin arm only

that multiple strains independently gained resistance against FQs through the same selective pressure.

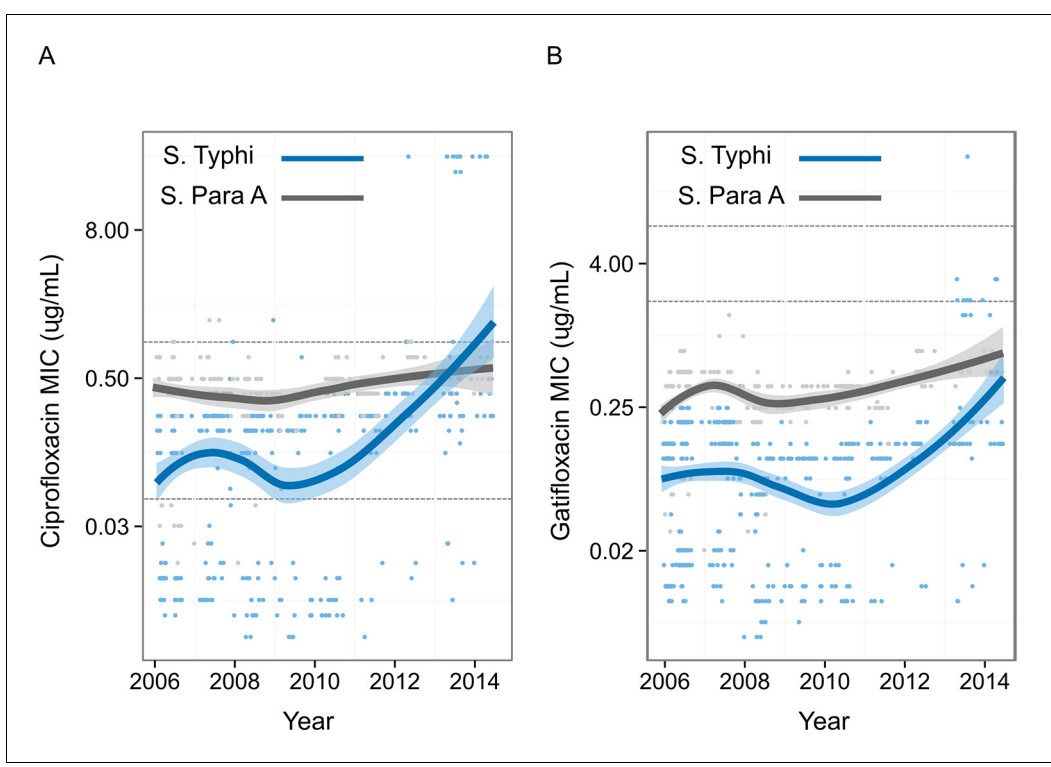

**Figure 3.** Minimum Inhibitory Concentrations of Nepali *Salmonella* Typhi and *Salmonella* Paratyphi against ciprofloxacin and gatifloxacin over ten years. Minimum Inhibitory Concentrations (μg/ml) for 568 Nepali *Salmonella* Typhi (blue) and 269 Nepali *Salmonella* Paratyphi A (grey) against (**A**) ciprofloxacin and (**B**) gatifloxacin collected from four randomised controlled trials conducted between 2005–2014 at Patan Hospital in Kathmandu, Nepal (***Pandit et al., 2007***; ***Koirala et al., 2013***; ***Arjyal et al., 2011***). The smoothed line derived from the generalized additive model showing a non-linear increase in Minimum Inhibitory Concentrations over time, with shading representing the 95% confidence interval. Lower and upper horizontal lines represent the current CLSI cut-offs for susceptible/intermediate and intermediate/resistant, respectively (***CLSI, 2012***).

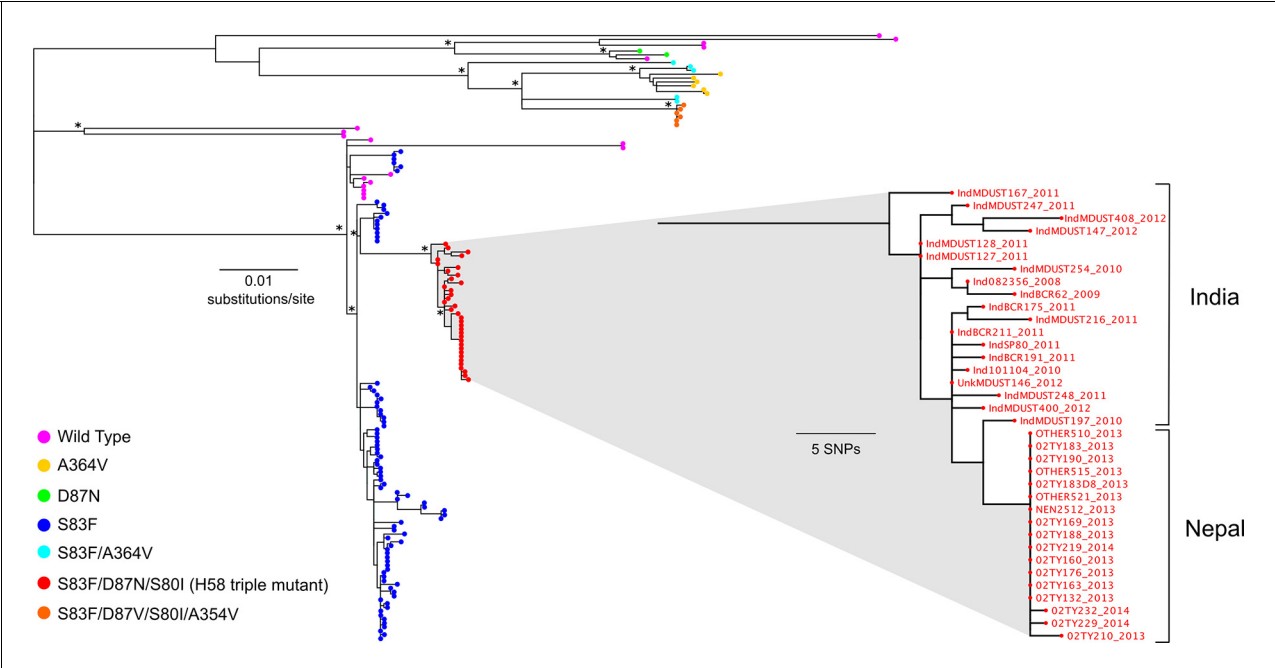

**Figure 4.** The phylogenetic structure of fluoroquinolone resistant *Salmonella* Typhi in a regional context. Maximum likelihood phylogeny based on core-genome SNPs of 136 (78 from the RCT) *Salmonella* Typhi isolates from Nepal and neighbouring India (*Supplementary file 1*). Main tree shows the overall phylogenetic structure and the presence of specific combinations of mutations in *gyrA* (S83F, D87V and D87N), *parC* (S80I) and *parE* (A364V). The inset shows a magnified view of the fluoroquinolone-resistant *Salmonella* Typhi H58 triple mutants from Nepal and their close association with similarly fluoroquinolone-resistant *Salmonella* Typhi H58 triple mutants from India (*Wong et al., 2015*). The scale bar on the primary tree indicates the number of substitutions per variable site, while that in the inset indicates genetic distance in number of SNPs (see methods). Asterisks indicate ≥85% bootstrap support at nodes of interest.

Appropriate antimicrobial therapy is critical in the treatment of enteric fever, as effective drugs curtail symptoms and prevent life threatening complications. Our data has substantial repercussions for enteric fever treatment, and we advocate that FQs should no longer be used for empirical enteric fever therapy on the Indian subcontinent, as we predict these strains are likely to be widespread and are associated with poor outcomes with FQ therapy. Notably, in the RCT from which these data were derived we used the newer generation FQ, gatifloxacin, which binds to a different location on the DNA gyrase than the older FQs and is not as susceptible to the common resistance mutations (*Lu et al., 1999*). The isolates in this study were not generally resistant to gatifloxacin according to the current CLSI guidelines for Enterobacteriaceae (*CLSI, 2012*); we suggest that these guidelines be modified specifically for *S.* Typhi to reflect these new clinical data. We additionally propose that *S.* Typhi genotyping, mapping and susceptibility testing is performed routinely and rapidly in reference laboratories inside and outside of South Asia to monitor the international spread of these strains and ensure the provision of alternative efficacious therapies to returning travellers (*Lee et al., 2013*; *García-Fernández et al., 2015*). In cases of infection with these FQ-resistant isolates, we suggest that ceftriaxone and azithromycin be used as alternatives, and do not currently recommend a return to the use of first-line drugs without contemporary data on treatment outcome. Whilst none of the isolates in this study were MDR, we predict a rapid return of MDR strains if there is a hasty return to older first-line alternatives.

This study has limitations. First, the clinical data was collected from one study in a single location, thus limiting utility outside this setting. Second, the overall sample size (and the gatifloxacin group subsampling) of those with culture-positive *S.* Typhi-associated enteric fever was relatively small, and the analysis presented here was performed in a *post hoc* manner. Notwithstanding these limitations, we were able to show a highly significant association between disease outcome and susceptibility

profile of the infecting organism. Further, by using WGS, we were able to pinpoint causative mutations, identify the subclade responsible for treatment failure and relate these strains to other isolates circulating outside Nepal in other parts of South Asia. The methodologies presented here, in which clinical outcome data are combined with genome sequences and antimicrobial susceptibility data, should become the gold standard for informing empiric treatment for all invasive bacterial infections and understanding the role of bacterial genotype and resistance profile on disease outcome for other bacterial infections. No other combination of methodologies would provide the granularity of data required to understand the epidemiology and clinical impact of this emergent strain in detail.

In conclusion, our data, for the first time, show a significant association between *S*. Typhi genotype, antimicrobial susceptibility and disease outcome for those treated with gatifloxacin in a cohort of Nepali enteric fever patients. A FQ-resistant variant of Typhi H58 has emerged in Nepal and is associated with the clinical failure of FQs. Our data suggest these isolates are likely widespread in the subcontinent and FQs should not be recommended for empirical enteric fever therapy in this setting.

## Materials and methods

### Study design and setting

The RCT from which the organisms and corresponding clinical data originated for these analyses was conducted at Patan Hospital and the Civil Hospital in the Lalitpur area of Kathmandu, Nepal, between 2011 and 2014, as described previously (*Arjyal et al., 2016*). The trial was registered at www.clinicaltrials.gov (ISRCTN63006567). Briefly, patients were randomly assigned to seven days of treatment with either oral gatifloxacin (400 mg tablets, Square Pharmaceuticals Limited, Bangladesh) at a dose of 10 mg/kg once daily or intravenous ceftriaxone (Powercef, 1000mg injection vial, Wockhardt Ltd, India), injected over 10 min at a dose of 60 mg/kg up to a maximum of two grams (aged 2 to 13 years) or two grams ($\geq$14 years) once daily.

A detailed description of the RCT from which these data were generated has been previously published (*Arjyal et al., 2016*). The primary endpoint was a composite of treatment failure, defined as the occurrence of at least one of the following events: fever clearance time (FCT) (time from the first dose of a study drug until the temperature dropped to $\leq$37·5℃ and remained there for at least two days) more than seven days post-treatment initiation; requirement for rescue treatment as judged by the treating physician; blood culture positivity for *S*. Typhi or *S*. Paratyphi on day eight of treatment (microbiological failure); culture-confirmed or syndromic enteric fever relapse within 28 days of initiation of treatment; and the development of any enteric fever-related complication (e.g. clinically significant bleeding, fall in the Glasgow Coma Score, perforation of the gastrointestinal tract and hospital admission) within 28 days after the initiation of treatment. Time to treatment failure was defined as the time from the first dose of treatment until the date of the earliest failure event. FCTs were calculated electronically using twice-daily recorded temperatures and treated as interval-censored outcomes. Patients without fever clearance or relapse, respectively, were censored at the time of their last follow-up visit (additional details regarding study procedures can be found in Arjyal *at al.* 2016 (*Arjyal et al., 2016*)).

Blood (3 ml if aged $\leq$14 years; 8 ml if aged $\geq$14 years) was taken from all patients for bacterial culture on enrolment. Adult blood samples were inoculated into media containing tryptone soya broth and sodium polyanethol sulphonate, up to a total volume of 50 mL. Bactec Peds Plus culture bottles (Becton Dickinson, New Jersey, USA) were used for paediatric blood samples. Culture results were reported for up to seven days, positive bottles were subcultured onto blood, chocolate and MacConkey agar and presumptive *Salmonella* colonies were identified using standard biochemical tests and serotype-specific antisera (Murex Biotech, Dartford, England). Antimicrobial susceptibility testing was performed by the modified Bauer-Kirby disc diffusion method with zone size interpretation based on CLSI guidelines (*CLSI, 2012*). Etests were used to determine MICs, following the manufacturer's recommendations (bioMérieux, France). Ciprofloxacin MICs were used to categorise *S*. Typhi isolates as susceptible ($\leq$0.06 µg/mL), intermediate (0.12–0.5 µg/mL) and resistant ($\geq$1 µg/mL) following CLSI guidelines (*CLSI, 2012*).

## Whole genome sequencing and analysis

Genomic DNA from Nepali *S.* Typhi organisms originating from this RCT (78 isolates) was extracted using the Wizard Genomic DNA Extraction Kit (Promega, Wisconsin, USA) (*Supplementary file 1*) (*Karkey et al., 2013*). Two µg of genomic DNA was subjected to WGS on an Illumina Miseq platform, following the manufacturer's recommendations to generate 250bp/100bp paired-end reads. All reads were mapped to the reference sequence of *S.* Typhi CT18 (accession no: AL515582) using SMALT (version 0.7.4). Candidate single nucleotide polymorphisms (SNPs) were called against the reference sequence using SAMtools (*Li et al., 2009*) and filtered with a minimal phred quality of 30 and a quality cut-off of 0.75. The allele at each locus in each isolate was determined by reference to the consensus base in that genome, using *samtools mpileup* and removing low confidence alleles with consensus base quality ≤20, read depth ≤5 or a heterozygous base call. SNPs called in phage regions, repetitive sequences or recombinant regions were excluded, (*Wong et al., 2015*) resulting in a final set of 1,607 chromosomal SNPs. Strains belonging to haplotype H58 were defined by the SNP *glpA*-C1047T (position 2348902 in *S.* Typhi CT18, BiP33) (*Emary et al., 2012*; *Holt et al., 2008*; *Parkhill et al., 2001*).

A maximum likelihood (ML) phylogeny was estimated using a 1440 SNP alignment of the 78 RCT isolates in RAxML (version 7.8.6) with the generalized time-reversible substitution model (GTR) and a gamma distribution, with support for the phylogeny assessed via 1000 bootstrap replicates. The alignment was then compared to a global *S.* Typhi sequence database, with a particular focus on identifying sequences with a mutational profile suggestive of shared ancestry with a divergent H58 clade identified in the previous phylogeny. A secondary ML phylogenetic tree was then inferred from the SNP alignment of the 136 Nepali Typhi along with 19 recently described Typhi H58 with the aforementioned mutational profile, using the same parameters as above (1642 SNPs; *Supplementary file 1*) (*Wong et al., 2015*). Raw sequence data are available in the European Nucleotide Archive (ENA) (*Supplementary file 1*).

## Statistical analysis

Comparison of baseline characteristics within patient groups, stratified by the H58 status or susceptibility category of their corresponding *S.* Typhi isolates was performed using the Kruskal Wallis test for continuous variables and Fisher's exact test for categorical variables. Time to treatment failure was analysed using Firth's penalized maximum likelihood bias reduction method for Cox regression as a solution for the non-convergence of likelihood function in the case of zero event counts in subgroups (*Firth, 1993*). For comparisons between treatment arms, H58 status, or ciprofloxacin susceptibility group, the model included treatment arm, H58 status, or susceptibility group as a single covariate. Confidence intervals (CI) and *p*-values were calculated by profile-penalized likelihood. FCT was analysed as an interval-censored outcome, i.e. as the time interval from the last febrile temperature assessment until the first afebrile assessment, using parametric Weibull accelerated failure time models (*Kalbfleisch and Prentice, 2002*). Median and inter-quartile range (IQR) FCT calculations for subgroups were based on models for each subgroup separately. Acceleration factors were based on models that included treatment arm as the only covariate. The non-parametric maximum likelihood estimator (NPMLE) was used to visualize the distribution of FCT between groups. Heterogeneity between subgroups was tested with models that included an interaction between treatment arm and the sub-grouping variable. To study the emergence of FQ resistance, data from previous enteric fever trials from 2005–2014 (*Pandit et al., 2007*; *Koirala et al., 2013*; *Arjyal et al., 2011*) was pooled and generalized additive models (GAM) were used to examine potential non-linear trends of ciprofloxacin and gatifloxacin MICs over time. All analyses were performed using R software version 3.2.2 (*Team, 2012*).

## Acknowledgements

This project was funded by the Wellcome Trust of Great Britain (106158/Z/14/Z). SB is a Sir Henry Dale Fellow, jointly funded by the Wellcome Trust and the Royal Society (100087/Z/12/Z). KEH is supported by fellowship #1061409 from the NHMRC of Australia. CD was funded by the Li Ka Shing Foundation Global Health Programme at the University of Oxford. DTP and AK are funded as leadership fellows through the Oak Foundation. The funders had no role in study design, data collection

and analysis, decision to publish, or preparation of the manuscript. We wish to acknowledge Mangal Rawal, Sumi Munankarmi, Bibek Karki, Radheshyam KC, Sudeep Dhoj Thapa, Rabin Gautam, Priyanka Tiwari, Manisha Risal, Surendra Shrestha, Balmukunda Neupane, Nabraj Regmi, Krishna Prajapati, Bimal Thapa and the trial monitors Nguyen Thi Phuong Dung and Nguyen Thi Thanh Thuy for their assistance in conducting this trial.

## Additional information

### Funding

| Funder | Grant reference number | Author |
|---|---|---|
| Wellcome Trust | 100087/Z/12/Z | Stephen Baker |
| Royal Society | 100087/Z/12/Z | Stephen Baker |
| National Health and Medical Research Council | 1061409 | Kathryn E Holt |
| Li Ka Shing Foundation | | Christiane Dolecek |
| Wellcome Trust | 106158/Z/14/Z | Kathryn E Holt<br>Gordon Dougan<br>Christiane Dolecek<br>Buddha Basnyat<br>Stephen Baker |
| The Oak Foundation | OCAY-15-547 | Duy Pham Thanh<br>Abhilasha Karkey<br>Stephen Baker |

The funders had no role in study design, data collection and interpretation, or the decision to submit the work for publication.

### Author contributions
DPT, Acquisition of data, Analysis and interpretation of data, Drafting or revising the article; AK, SD, VW, NTVT, PVV, THT, DP, Acquisition of data, Contributed unpublished essential data or reagents; NHT, CNT, MAR, KEH, Analysis and interpretation of data, Drafting or revising the article; AA, BB, Conception and design, Acquisition of data; AP, SKS, DG, Recruited patients into study and collected clinical data, Acquisition of data; CMP, GD, GET, Conception and design, Drafting or revising the article; MW, Analysis and interpretation of data, Contributed unpublished essential data or reagents; CD, Conception and design, Acquisition of data, Contributed unpublished essential data or reagents; SB, Conception and design, Acquisition of data, Analysis and interpretation of data, Drafting or revising the article

### Author ORCIDs
Maia A Rabaa, http://orcid.org/0000-0003-0529-2228

### Ethics
Clinical trial Registry: ISRCTN. Registration ID: ISRCTN63006567.
Human subjects: This study was performed following the principles of the declaration of Helsinki. Written informed consent to participate in all studies from Nepal contributing data for this analysis was required from all patients. For those aged <18 years, written informed consent was obtained from a parent or an adult guardian. The protocol was reviewed and approved by the Ethics Committee of the Nepal Health Research Council (NHRC) and the Oxford Tropical Research Ethics Committee (OxTREC) UK.

## Additional files

### Supplementary files
• Supplementary file 1. Table of *Salmonella* Typhi isolates and their corresponding sequencing metadata used in this study.

• Supplementary file 2. (A) Table of baseline characteristics by *Salmonella* Typhi lineage. (B) Table of baseline characteristics grouped by *Salmonella* Typhi ciprofloxacin susceptibility. (C) Table of treatment failure in detail by *Salmonella* Typhi lineage in the gatifloxacin treatment group. (D) Table of treatment failure in detail by ciprofloxacin susceptibility in the gatifloxacin treatment group.

## Major datasets

The following dataset was generated:

| Author(s) | Year | Dataset title | Dataset URL | Database, license, and accessibility information |
|---|---|---|---|---|
| Duy Pham Thanh, Stephen Baker, Kathryn E Holt, Gordon Dougan, Vanessa Wong | 2015 | Sequence data | http://www.ebi.ac.uk/ena/data/view/PRJEB10959 | PRJEB10959 |

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
