## [Decision Letter]

Thank you for submitting your work entitled "A novel ciprofloxacin-resistant subclade of H58 *Salmonella Typhi* is associated with fluoroquinolone treatment failure" for consideration by *eLife*. Your article has been reviewed by three peer reviewers, one of whom, Brent House, has agreed to reveal his identity, and the evaluation has been overseen by Prabhat Jha as the Reviewing and Senior Editor.

The reviewers have discussed the reviews with one another and the editor has drafted this decision to help you prepare a revised submission.

Summary:

We believe this is a strong paper that could be published in *eLife*. To do so we request that you consider one (reasonably minor) revision and address also some minor points.

Please address these points in a revised manuscript with a point-by-point reply to the key revisions/minor points for us to consider further.

Essential revisions:

One weakness we observed with the article is that there is a huge amount of information from the clinical trial (Arjyal et al.et al., 2016) that is repeated in the Methods section, subheading “Study design and setting”, and this appears rather redundant.

*Reviewer #1:*

Considering the limitations, this research project was well designed and the manuscript well written. I believe the statistical analyses, Discussion and Conclusion were all accurate and reasonable.

*Reviewer #2:*

This is a very well written article with profound outcomes for the management of typhoid fever in South East Asia and other regions where resistance to fluoroquinolones has emerged as a major concern.

The detection of *S. Typhi* isolates within H58 subclade that have triple mutations in the QRDR is a major finding that explains the apparent treatment failure observed in the recently concluded clinical trial quoted by authors (Arjyal et al., 2016) and earlier in studies in India, and this will inform clinical management where similar strains are isolated in the region and elsewhere.

One weakness I observed with the article is that there is a huge amount of information from the clinical trial (Arjyal et al., 2016), that is repeated in the Methods section, subheading “Study design and setting” and this appears rather redundant.

*Reviewer #3:*

This is a well written and, I believe, significant report. It grows out of a large clinical trial performed in the Kathmandu Valley in Nepal that was comparing gatifloxacin with ceftriaxone in the treatment of individuals with enteric (typhoid) fever.

During the study period, there was emergence of a highly resistant strain of *Salmonella typhi*, the cause of typhoid fever (requiring the DSMB to halt the study because of the high failure rate). The researchers were thus in a unique situation. They had detailed clinical information, as well as the isolates, and were able to perform whole genome sequencing of the collected strains. They were then able to perform a detailed phylogenetic analysis, correlating that with the clinical information, specifically fourth generation fluoroquinolone failure. The main finding of the study is that the authors identified a subclade of the H58 lineage that had not only the DNA (*gyrA*) S83F mutation, but two additional mutations D87N (*gyrA*) and S80I in topoisomerase *parC*. The subclade that had triple mutations was associated with much higher MICs to ciprofloxacin, decreased fever clearance time, and clinical failure. They compared this subclade to a repository and identified relationships with isolates circulating in India prior to the time of the study. They hypothesize that the subclade was introduced into Nepal from India and is spreading.

I believe the work is significant for a number of reasons. It is to the best of my knowledge one of the most detailed studies correlating whole genome analysis to clinical outcome in the typhoid field (a human-restricted infection of high global importance; since it is human-restricted among impoverished people in resource-limited areas, it is notoriously hard to investigate), especially pertaining to the specific development and evolution of drug resistance over time. The work is not only scientifically informative but clinically extremely useful. The authors are correct that their results strongly suggest that even advanced fluoroquinolone antibiotics can no longer be used for the treatment of individuals with typhoid fever in the Indian subcontinent, and that this subclade may rapidly spread globally (the H58 clade has globally spread very rapidly, replacing previous clades, and appears to have a fitness advantage). The removal of fluoroquinolones from the therapeutic armamentarium cannot be overstated. It is the life blood of treating patients with typhoid (oral, cheap, effective). But the emergence of this H58 subclade now removes FQ as a weapon class in South Asia (and probably shortly globally given the H58 provenance); requiring either falling back on intravenous agents (not practical/optimal/feasible for many patients with typhoid) or azithromycin, the last oral chess piece on the table. It is in part due to this context, that I believe the work has additional unique global significance.

I have no major concerns with the manuscript.

---

## [Author Response]

*Essential revisions:*

One weakness we observed with the article is that there is a huge amount of information from the clinical trial (Arjyal et al.et al., 2016) that is repeated in the Methods section, subheading “Study design and setting”, and this appears rather redundant.

Yes we agree and this required to be added at an earlier stage, we have referenced the majority of this but left in the important study design features to ensure the paper can be read without reference to the RCT.